# Comparative Effectiveness of Various Multi-Antigen Vaccines in Controlling *Campylobacter jejuni* in Broiler Chickens

**DOI:** 10.3390/vaccines12080908

**Published:** 2024-08-10

**Authors:** Mostafa Naguib, Shreeya Sharma, Abigail Schneider, Sarah Wehmueller, Khaled Abdelaziz

**Affiliations:** 1Department of Animal and Veterinary Science, Clemson University, Clemson, SC 29634, USA; mnaguib@clemson.edu (M.N.); shreeys@clemson.edu (S.S.); arschne@g.clemson.edu (A.S.);; 2Department of Poultry Diseases, Faculty of Veterinary Medicine, Cairo University, Cairo 12211, Egypt; 3Clemson University School of Health Research (CUSHR), Clemson, SC 29634, USA

**Keywords:** chickens, *Campylobacter*, CpG ODN, vaccine, outer membrane protein, in ovo

## Abstract

This study was undertaken to evaluate and compare the efficacy of different multi-antigen vaccines, including heat-inactivated, whole lysate, and subunit (outer membrane proteins [OMPs]) *C. jejuni* vaccines along with the immunostimulant CpG ODN in controlling *Campylobacter* colonization in chickens. In the first trial, 125 μg of *C. jejuni* OMPs and 50 μg of CpG ODN were administered individually or in combination, either in ovo to chick embryos or subcutaneously (SC) to one-day-old chicks. In the second trial, different concentrations of *C. jejuni* antigens (heat-killed, whole lysate, and OMPs) were administered SC to one-day-old chicks. The results of the first trial revealed that SC immunization with the combination of CpG ODN and *C. jejuni* OMPs elevated interferon (IFN)-γ, interleukin (IL)-1β, and IL-13 gene expression in the spleen, significantly increased serum IgM and IgY antibody levels, and reduced cecal *C. jejuni* counts by approximately 1.2 log_10_. In contrast, in ovo immunization did not elicit immune responses or confer protection against *Campylobacter*. The results of the second trial showed that SC immunization with *C. jejuni* whole lysate or 200 μg OMPs reduced *C. jejuni* counts by approximately 1.4 and 1.1 log_10_, respectively. In conclusion, *C. jejuni* lysate and OMPs are promising vaccine antigens for reducing *Campylobacter* colonization in chickens.

## 1. Introduction

*Campylobacter* is the leading cause of diarrheal disease in the US, accounting for approximately 1.5 million cases of human campylobacteriosis annually [1]. Thermophilic *Campylobacter*, mainly *C. jejuni* and *C. coli*, commonly colonize wild birds’ guts and domestic fowl (e.g., chickens, turkeys, ducks, and geese) [2,3]. *C. jejuni* colonizes the intestines of chickens during the second or third week of age, with up to 10^8^ colony-forming units (CFUs)/g found in each gram of intestinal content in infected chickens [4,5,6]. Transmission of this organism to humans usually occurs through contact with farm animals or by consuming their contaminated products, with poultry products being the primary source of infection in most cases [4]. Therefore, it is imperative to implement pre- and post-harvest control measures to decrease the *Campylobacter* load on poultry carcasses, thereby mitigating the risk of *Campylobacter* transmission to humans.

Several pre-harvest approaches have been applied to control *Campylobacter* colonization rates in broiler chickens, including biosecurity practices [5], the use of feed additives, such as prebiotics, probiotics, synbiotics, bacteriophages, organic acids, short-chain fatty acids, and essential oils [6,7,8], and drinking water sanitation [9]. While these interventions have shown potential in reducing *C. jejuni* loads in chicken intestines, none have completely eliminated its colonization, necessitating exploring more efficacious strategies for on-farm control of *Campylobacter*. Indeed, vaccination is considered one of the potential measures to control *Campylobacter* infection in chickens. Yet, none of the vaccines developed by various research groups offer “complete” protection against this bacterium in chickens [10,11,12,13,14].

The challenges in the development of effective vaccination against *Campylobacter* lie in the unsuccessful identification of a novel, highly conserved immunogenic protein that can induce cross-protective immunity against different strains of *C. jejuni*, along with the lack of targeted delivery methods for these antigens to mucosal immune inductive sites [15]. In this context, various multi-antigens vaccines, including killed, live attenuated, subunit, and recombinant vaccines, have been explored with differing levels of success [6,16]. Despite their multi-antigen composition, the ineffectiveness of these vaccines in providing adequate protection against *Campylobacter* was attributed to their lack of effective immunogenic proteins and the unsuccessful experimental setup, such as the use of suboptimal dosages and inefficient vaccination routes.

Among these vaccines, the outer membrane proteins (OMPs) subunit vaccine has shown promise for controlling bacterial foodborne pathogens in chickens, including *Salmonella* and *Campylobacter* [11,12,17]. For instance, Annamalai and colleagues have demonstrated that subcutaneous (SC) administration of a crude mixture of *C. jejuni* OMPs induced systemic protective antibody responses associated with a reduction in *C. jejuni* colonization in broiler chickens [11]. However, the SC route is impractical for mass administration in poultry production, necessitating further investigation into more suitable methods. While mucosal routes are appropriate for mass vaccination, oral delivery of vaccines remains challenging due to the potential degradation of vaccine antigens and adjuvants by gastric juice and digestive enzymes [13,14]. On the other hand, the in ovo route offers a reliable, cost-effective, and efficient approach for mass immunization, making it an attractive option for vaccinating thousands of birds at hatcheries, in contrast to laborious post-hatching vaccination procedures [18]. Additionally, vaccine antigens delivered to chick embryos before hatching are not diluted by ingested food and have limited exposure to digestive enzymes, thereby prolonging their contact with mucosal tissues.

In contrast to live attenuated vaccines, it is widely agreed that inactivated and subunit vaccines require the incorporation of natural or synthetic adjuvants to enhance their immunogenicity [19]. Among these adjuvants, CpG ODN, a synthetic single-stranded oligodeoxynucleotide containing unmethylated CpG motifs, has demonstrated potential both as a standalone antimicrobial agent and as a vaccine adjuvant [20,21]. CpG ODN is recognized by avian Toll-Like Receptor 21 (TLR21), which is analogous to TLR9 in mammals [22]. When CpG ODN binds to TLR21, it triggers intracellular signaling pathways that stimulate the secretion of immunomodulatory substances like cytokines, chemokines, and antimicrobial peptides, which in turn enhance protection against bacterial pathogens [22,23]. Recent studies have shown that in ovo administration of CpG ODN enhances chicks’ resistance to *Escherichia coli* and *Salmonella Typhimurium* infections [24]. However, the potential of in ovo-administered CpG ODN and *C. jejuni* OMPs to confer protection against *C. jejuni* has yet to be investigated. Therefore, this study was carried out to assess the efficacy of the in ovo administration of *C. jejuni* OMPs and CpG ODN, either individually or in combination, against *Campylobacter* in broiler chickens. A dose optimization trial was also conducted to evaluate and compare the protective effects of different dosages of various multi-antigen *Campylobacter* vaccines, including whole-cell (heat-killed bacteria and whole lysate) and subunit OMPs vaccines.

## 2. Materials and Methods

### 2.1. Preparation of Campylobacter Culture for Experimental Challenge

*C. jejuni* strain 81–176 was cultured as described previously [13], with minor modifications. Briefly, a loop of frozen *C. jejuni* glycerol stock was streaked onto Brain Heart Infusion (BHI) agar containing Preston *Campylobacter* Selective Supplement (Thermo Fisher Scientific, Rockford, IL, USA) and incubated for 24 h at 37 °C under microaerobic conditions of 10% CO_2_, 5% O_2_, and 85% N_2_. Subsequently, a few colonies were inoculated into 5 mL fresh BHI broth and incubated at 37 °C under microaerobic conditions. Following incubation for 24 h, 1 mL of the bacterial suspension was transferred to 100 mL of BHI broth and re-incubated at 37 °C for 40 h. Afterward, the bacterial suspension was centrifuged at 3500× *g* for 10 min, re-suspended in phosphate buffer saline (PBS; pH 7.4), and the concentration was estimated based on the optical density (OD) measured at 600 nm.

### 2.2. Vaccine Preparation

#### 2.2.1. CpG ODN

A phosphorothioate-based synthetic class B 2007 CpG ODN, purchased from Invivogen (San Diego, CA, USA), was reconstituted in endotoxin-free water and diluted to working quantities in PBS. For in ovo immunization, 0.1 mL containing 50 µg was injected into the amniotic fluid of the fertilized egg, whereas for SC immunization, 0.2 mL containing 50 µg CpG ODN was injected SC.

#### 2.2.2. Preparation of *Campylobacter* OMPs

The OMPs of *C. jejuni* 81–176 were extracted as described previously [25]. Briefly, *C. jejuni* was grown in 5–10 L of BHI broth following the abovementioned procedure. Following centrifugation for 10 min at 3000× *g*, the bacterial suspension was washed twice with distilled water. Four grams of packed cells were thoroughly mixed in 100 mL of 0.2 M glycine-hydrochloride buffer (pH 2.2) and stirred for 15 min at room temperature. The suspension was then centrifuged for 15 min at 11,000× *g*. Afterward, the supernatant was collected and neutralized with 1 N NaOH, and extensive dialysis was performed overnight at 4 °C against deionized water. The protein concentration of the OMPs was quantified using the BCA Protein Assay Kit (Thermo Fisher Scientific, Rockford, IL, USA) and stored at −80 °C until use. Protein separation was confirmed by using SDS-PAGE and Coomassie Blue staining.

#### 2.2.3. Preparation of the Killed *Campylobacter* Vaccine

*C. jejuni* was grown in BHI broth, as described above. After washing twice with PBS, the bacterial pellet was resuspended in PBS, and OD was measured. Subsequently, bacterial concentration was adjusted to 5 × 10^7^ and 5 × 10^8^ colony forming units (CFUs) per mL of PBS. Subsequently, bacterial suspension was heat-inactivated at 65 °C for 30 min. An aliquot (100 µL) of the bacterial suspension was streaked onto BHI agar and incubated under microaerobic conditions at 37 °C for 48 h to ensure the complete killing of bacterial cells. No bacterial growth was detected on the agar plate.

#### 2.2.4. Preparation of the *Campylobacter* Lysate

*C. jejuni* was grown and resuspended in PBS as described above. The bacterial suspension was subsequently sonicated on ice (twelve 15-second pulses interrupted with 30-second pulses). To ensure complete sonication of the live bacteria, 100 µL of the bacterial suspension was streaked onto BHI agar and incubated under microaerobic conditions at 37 °C for 48 h. No bacterial growth was detected on the agar plate. The protein concentration of the lysate was quantified using the BCA Protein Assay Kit (Thermo Fisher Scientific, Rockford, IL, USA) and served as a biomarker for dosage determination. 

### 2.3. Egg Incubation and Chicken Housing

Commercial fertilized Ross 308 broiler eggs were obtained from a commercial hatchery (Fieldale Farms Corporation, Baldwin, GA, USA) and incubated in a sanitized egg incubator (GQF Manufacturing Company Inc., Savannah, GA, USA) at the Morgan Poultry Center of Clemson University until hatching. Hatched chicks were transferred to the Godley-Snell Facility of Clemson University, where they were fed antibiotic- and additives-free diets ad libitum. All procedures were approved by the Institutional Animal Care and Use Committee (IACUC) at Clemson University (AUP 2022-0411)

### 2.4. Experimental Design

#### 2.4.1. First Trial

Following incubation and egg candling, 272 embryonated eggs were randomly divided into eight groups, as depicted in Table 1. On the embryonic day (ED)18, 136 eggs were disinfected with 70% ethanol and allocated into four groups (G1–G4), each containing 34 eggs. Afterward, the eggshell was pierced using a 23-gauge needle, and 50 µg CpG ODN in 100 µL PBS or 125 µg *C. jejuni* OMPs in 100 µL PBS or their combination (50 µL of PBS containing 50 µg CpG ODN mixed thoroughly with 50 µL PBS containing 125 µg OMPs) were injected into the amniotic fluid of the fertilized eggs. The eggs in G4 were injected with PBS and served as a negative control group. The concentrations of CpG ODN and *C. jejuni* OMPs used in this study were chosen for their established immunostimulatory capabilities and demonstrated effectiveness in reducing *Campylobacter* counts in previous research [11,24].

Following inoculation, the eggs were transferred to the hatchery. On ED 19, 20, and 21 (days one, two, and three post-immunization [PI]), eight birds per group were euthanized, and the spleen and bursa of Fabricius were collected for gene expression analysis. The remaining eggs continued their incubation until hatching. The hatchability percentage was 94% in the in ovo immunized groups and 85% in the non-immunized groups, indicating that the injected vaccines did not impact the hatchability rate. Hatched chicks in G1-8 were housed in separate pens. 

On the first day post-hatch, the non-immunized chicks (*n* = 136) were randomly divided into four groups (G5–G8), each containing 34 birds. As depicted in Table 1, one-day-old chicks were injected SC with 50 µg CpG ODN in 200 µL PBS or 125 µg *C. jejuni* OMPs in 200 µL PBS or their combination (100 µL of PBS containing 50 µg mixed thoroughly with 100 µL PBS containing 125 µg OMPs) or PBS. On the second, third, and fourth day of age (days one, two, and three PI), eight birds per group were euthanized, and the spleen and bursa of Fabricius were collected for gene expression analysis.

At one week of age, chicks in groups 1–8 received the booster dose of the respective vaccines, administered orally for groups 1–4 and SC for groups 5–8. At two weeks of age, all groups were challenged orally with 10^7^ CFUs of *C. jejuni* strain 81–176 in 1 mL PBS. Blood samples were collected weekly from all groups (G1-8), starting from the first week of age through the fifth week of age, with the sera subsequently separated for measuring the antibody (Ab) levels. On day 35 of age, all chickens were euthanized, and cecal contents were collected to enumerate the *C. jejuni* colony count (Figure 1).

#### 2.4.2. Second Trial

Eighty one-day-old chicks were randomly divided into eight groups (G1–G8), each containing ten chicks. As depicted in Table 2, chicks were immunized SC with different *C. jejuni* antigens (the heat-killed or whole lysate or OMPs) or PBS at day one of age. On day 14 of age, chicks received a booster dose of the respective vaccine SC. On day 15 of age, all groups were orally challenged with 10^7^ CFUs of *C. jejuni* strain 81-176 in 1 mL PBS. Blood samples were collected weekly from all groups (G1-8), starting the first week of age through the fifth week of age, with the sera subsequently separated for measuring the antibody levels. All chickens were euthanized on day 35 of age, and cecal contents were collected for enumeration of *C. jejuni* colony count (Figure 2). 

### 2.5. RNA Extraction and Complementary DNA (cDNA) Synthesis

The bursa of Fabricius and spleen tissues were homogenized using Bead Ruptor Elite (Omni International, GA, USA) and the RNA was extracted using TRIzol (Invitrogen, Carlsbad, CA, USA), according to the manufacturer’s protocol. Total RNA was treated with DNase (DNA-free kit, Invitrogen, Carlsbad, CA, USA) to eliminate the genomic DNA. The quality and concentration of RNA were measured by a Nanodrop One spectrophotometer (Thermo Scientific, Greenville County, SC, USA). Reverse transcription to cDNA was carried out using the SuperscriptII First-Strand Synthesis kit (Invitrogen, Carlsbad, CA, USA) and oligo-dT primers (Thermofisher Scientific, Greenville County, SC, USA), following the manufacturer’s protocol. The cDNA was diluted 1:10 in nuclease-free water.

### 2.6. Quantitative Real-Time PCR (RT-qPCR)

RT-qPCR was performed using the LightCycler480 system (Roche Diagnostics), as previously described [26]. In summary, the PCR master mix contained 3 µL of nuclease-free water, 1 µL of forward and 1 µL of reverse primers (10 µM), and 10 µL of PowerTrack SYBR Green Master Mix (ThermoFisher Scientific, Baltics UAB, Vilnius, Lithuania). The total volume of each reaction was 20 µL consisting of 5 µL of cDNA and 15 µL of the master mix. The RT-qPCR cycling parameters comprised a 95 °C denaturation step, 45 cycles of amplification (95 °C for 10 s), annealing (optimal temperature for each primer is provided in Table 3), and extension (72 °C for 10 s). To make the melting curve, heating to 95 °C for 10 s, cooling to 65 °C for 1 min, and heating again to 97 °C was performed. In this investigation, all primers used were synthesized by MilliporeSigma (Burlington, MA, USA). Roche LightCycler 480 software was used to calculate the expression of the target genes relative to the reference gene (β-actin), employing the 2^−ΔΔCT^ method as described earlier [27], with qPCR efficiency ranging from 95% to 100%.

### 2.7. Enzyme-Linked Immunosorbent Assay (ELISA) for Measuring Serum IgY and IgM Antibody Levels

Serum IgY and IgM Ab levels were measured as previously described [35]. Briefly, Maxisorp 96 well plates (Thermo Fisher Scientific, Rochester, NY, USA) were coated with OMPs of *C. jejuni* (0.39 µg/100 µL) in PBS (pH 7.4) and incubated at 37 °C for two hours. The plates were washed four times with the washing buffer (PBS containing 0.05% Tween 20), then blocked with blocking buffer containing 0.5% pig gelatin (Sigma, St. Louis, MO, USA) and 0.05% Tween 20 in PBS and incubated at 37 °C for one hour. Following blocking, 100 µL of sera diluted 1/10 in PBS containing 1.5% Tween-20 and 0.29 M NaCl were added to wells in duplicate, followed by a one-hour incubation at 37 °C. After the plates were washed four times with the washing buffer, 100 µL of goat-anti-chicken HRP-conjugated IgM (Invitrogen, USA) or IgY (Sigma, USA) antibodies were added at a dilution of 1/10,000 or 1/4000, respectively, and then incubated at 37 °C for 30 min. After washing the plates twice with the washing buffer, 100 µL of ABTS (2, 2′-azino-di (3-ethyl-benzthiazoline-6-sulfonate)) substrate (Life Technologies, Frederick, MD, USA) was added to wells. After incubation at room temperature for 30 min, the reaction was stopped by adding 1% sodium dodecyl sulfate (Bio-Rad, Hercules, CA, USA) to wells and the optical densities were evaluated at 405 nm. 

### 2.8. Enumeration of C. jejuni Colony Count

At 35 days of age, chickens in all groups were necropsied, cecal contents were collected, and ten-fold serial dilutions were performed for up to six dilutions in PBS. Each dilution was plated on BHI agar containing Preston *Campylobacter* Selective Supplement. Plates were incubated for 48 h at 37 °C under microaerobic conditions (85% N2, 10% CO_2_, and 5% O_2_). The cecal *C. jejuni* CFUs were quantified and presented as log_10_ *C. jejuni*/gram of cecal content.

### 2.9. Statistical Analysis

Data were analyzed using JMP^®^ Pro 17.1.0 (JMP, SAS Institute Inc., Cary, NC, USA), and graphs were created using GraphPad Prism V5.0 (GraphPad Software, San Diego, CA, USA). The Shapiro–Wilk test was used to assess the normality of data distribution. The impact of treatments on colony count, relative expression of immune genes, and Ab levels were assessed using both parametric and non-parametric tests. For normally distributed data, one-way ANOVA was used, followed by Tukey’s post hoc test to determine differences among the means of treatment groups. For non-normally distributed data, the Kruskal–Wallis test was used, followed by Dunn’s test. Data are presented as the mean of colony count, relative gene expression and Ab level ± standard error of the mean (SEM). *p* < 0.05 was considered significant for all statistical tests. The correlation between serum Ab levels at the fifth week of age and CFUs of *Campylobacter* was assessed using Pearson’s r correlation coefficient.

## 3. Results

### 3.1. First Trial

#### 3.1.1. The Effects of in Ovo and SC Administration of *C. jejuni* OMPs and CpG ODN on Cecal Colonization with *C. jejuni*

The number of *C. jejuni* CFUs per gram of cecal content varied significantly among the immunized and non-immunized groups (*p* < 0.05) (Figure 3). No significant reduction in *Campylobacter* counts was observed in the groups immunized in ovo with CpG ODN or *C. jejuni* OMPs or their combination compared to the PBS control group. On the other hand, SC immunization of chickens with the combination of 50 μg CpG ODN and 125 μg *C. jejuni* OMPs significantly reduced cecal colonization with *C. jejuni* by 1.2 log_10_ (*p* < 0.05). While SC immunization with 50 μg CpG ODN reduced *C. jejuni* colonization by 1 log_10_, immunization with 125 μg *C. jejuni* OMPs had no significant effect on the *Campylobacter* count. A subsequent study was conducted to determine if using different concentrations of *C. jejuni* OMPs and adjusting the timing of the secondary/booster vaccination would result in a greater reduction in *Campylobacter* count.

#### 3.1.2. The Effects of in Ovo and SC Administration of *C. jejuni* OMPs and CpG ODN on the Serum Ab Levels

##### Serum IgY Ab Levels

No significant differences in IgY Ab levels were observed among the immunized and control groups at the first and second weeks of age. However, by the third week of age, IgY Ab levels were significantly elevated only in the group immunized with the combination of CpG ODN and *C. jejuni* OMPs (*p* < 0.0001) and consistently increased until the fifth week of age, compared to the PBS control group (Figure 4). The level of IgY Ab levels was significantly higher in the group immunized SC with both CpG ODN and *C. jejuni* OMPs compared to the other immunized groups except at week four of age, where it was not significantly different from the group immunized in ovo with CpG ODN. No correlation (r = 0.3962, *p* = 0.145) was observed between the cecal *C. jejuni* CFUs and serum IgY antibody levels in the group immunized with a combination of 50 µg CpG ODN and 125 µg OMPs at the fifth week of age (Figure 5).

##### Serum IgM Ab Levels

Similar to the pattern observed with IgY antibody levels, chickens immunized SC with both 50 μg CpG ODN and 125 μg *C. jejuni* OMPs exhibited significantly higher IgM levels than the PBS control group, starting from the third week (*p* < 0.0001) through the fifth week of age (Figure 6). However, the groups immunized SC and in ovo with either 125 μg *C. jejuni* OMPs or CpG ODN alone did not show significant increases in IgM antibody levels at any time point.

A significant increase in IgM antibody levels was observed in the group immunized in ovo with both CpG ODN and *C. jejuni* OMPs only at the fourth week of age. Similar to the observation made for serum IgY, no correlation was observed between the groups with the low colony count and high IgM Ab titers at the fifth week of age (r = 0.4457, *p* = 0.1146) (Figure 7). 

#### 3.1.3. The Effects of in Ovo and SC Administration of *C. jejuni* OMPs and CpG ODN on Cytokine Gene Expression in the Spleen and Bursa of Fabricius

The expression levels of IFN-γ, interleukin (IL)-1β, IL-4, IL-10, IL-13, transforming growth factor (TGF)-β), and B cell activating factor (BAFF) were measured in the spleen and bursa of Fabricius during three consecutive days PI. Spleens of chicks immunized SC with 50 μg CpG ODN showed a significant increase in IFN-γ expression at 24 h PI (*p* < 0.05) (Figure 8). However, no significant differences in IFN-γ gene expression were observed at 48 and 72 h PI compared to the PBS control group. Spleens of chicks in the groups immunized SC with 50 μg CpG ODN and the combination of 50 μg CpG ODN and 125 μg *C. jejuni* OMPs exhibited significantly increased expression levels of IL-13 at 24 h compared to the PBS control group. A significant upregulation of IL-1β expression was observed in the group immunized SC with both CpG ODN and *C. jejuni* OMPs at 24 and 48 h PI (*p* < 0.05) (Figure 8). However, SC immunization with *C. jejuni* OMPs did not significantly alter the expression of IFN-γ, IL-13, and IL-1β at any time point when compared against the PBS control group. SC and in ovo immunization with *C. jejuni* OMPs alone did not significantly modulate the gene expression of IFN-γ, IL-13, and IL-1β at all time points compared to the PBS control group (Figure 8 and Figure 9). No significant alterations were observed in the expression levels of IL-4, IL-10, TGF-β, and BAFF in the spleen of all immunized groups at any time point. 

No significant changes were observed in the expression levels of all the genes measured in this study in the bursa of Fabricius of all immunized groups.

### 3.2. Second Trial

#### 3.2.1. The Effects of SC Administration of Various Concentrations of *C. jejuni* OMPs, Heat-Killed and Whole Lysate Vaccines on Cecal Colonization with *C. jejuni*


Immunizing chickens SC on the first day and second week of age with a low dose of *C. jejuni* lysate (21.5 µg) or low (50 μg) and high (200 μg) doses of *C. jejuni* OMPs significantly reduced cecal colonization with *C. jejuni* by 1.4 (*p* = 0.0004), 1 (*p* = 0.038), and 1.1 (*p* = 0.005) log_10_ per gram of content, respectively, compared to the PBS control group (Figure 10). However, a lower but not statistically significant reduction of cecal colonization by 0.8 log_10_ was observed in the group immunized with 125 μg OMPs. No significant reductions in *C. jejuni* colony counts were observed in the groups immunized with a high dose (43 μg) of lysate and both low (10^6^ CFUs) and high (10^7^ CFUs) doses of heat-killed *C. jejuni.*

#### 3.2.2. The Effects of SC Administration of Various Concentrations of *C. jejuni* OMPs, Heat-Killed and Whole Lysate Vaccines on the Serum Ab Levels

##### Serum IgY Ab Levels 

No significant changes in the levels of IgY Ab levels were observed among the immunized and the PBS control groups during the first three weeks of age (Figure 11). However, by the fourth week of age, IgY Ab levels were significantly elevated only in the group immunized with the low dose (50 µg) of OMPs and continued to rise until the fifth week of age, compared to the PBS control group (*p* < 0.0001). Significantly higher IgY Ab levels were observed in the group immunized with the high dose of *C. jejuni* lysate only at the fourth week of age (*p* < 0.0001) compared to the PBS control group. However, no significant changes in the levels of IgY Ab levels were observed in the other immunized groups compared to the PBS control group (Figure 11). 

##### Serum IgM Ab Levels 

No significant differences in the levels of IgM Ab levels were observed among the immunized and PBS control groups at all time points (Figure 12). 

## 4. Discussion

Reducing the *Campylobacter* load on chicken meat is crucial for preventing human infection with this pathogen [36]. A recent modeling study estimated that reducing cecal colonization with *Campylobacter* by 3 log_10_ units could lower the incidence of disease in humans by 58% [37]. Since no commercial vaccines are currently available, further research is needed to explore effective vaccination strategies to reduce *Campylobacter* burden in chickens.

*Campylobacter* establishes early colonization by the second or third week of age due to the limited capacity of maternally derived antibodies (MDA) to provide prolonged protection beyond the second week of age [12,38]. Thus, early vaccination is necessary to boost the chick’s resistance to *Campylobacter* infection.

Despite the incomplete development of the immune system in the chicken embryo, accumulating evidence indicates that in ovo administration of vaccines and immunostimulants induces immune responses in lymphoid organs, including the spleen and bursa of Fabricius [39,40] and confer protection against viral and bacterial diseases [41,42,43,44,45]. Indeed, in ovo vaccination is currently being used worldwide for protection against viral diseases, including Marek’s disease [42] and infectious bursal disease [46]. In the context of foodborne pathogens, in ovo administration of TLR21 ligand (CpG ODN) has been shown to elicit a robust immune response and enhance chickens’ resistance against infection with *E. coli* and *Salmonella* species [21,47]. Along similar lines, in ovo delivery of *Salmonella* OMPs-loaded nanoparticles has been shown to induce antigen-specific immune response associated with a reduction in *Salmonella* colonization [48]. 

The effectiveness of in ovo vaccination can be attributed to the rapid and concurrent activation of immune responses in mucosal surfaces and lymphoid organs since these antigens can be readily uptaken from the amniotic fluid by multiple routes, including oral, respiratory and cloacal routes [19,20]. Despite the effectiveness of in ovo delivered CpG ODN and bacterial OMPs in modulating the immune system of chick embryos [20,49] and providing protection against *E. coli* and *Salmonella* [21], no studies have assessed their protective effects against *Campylobacter* infection in chickens when administered in ovo. Therefore, the first goal of this study was to evaluate and compare the protective efficacy of *C. jejuni* OMPs and CpG ODN, either individually or in combination, against *Campylobacter* infection in broiler chickens when administered either in ovo to chick embryos or SC to hatched chicks.

While Annamalai and colleagues demonstrated a significant reduction in *Campylobacter* count in chickens immunized SC with 125 µg of *C. jejuni* OMPs [11], this concentration of OMPs did not exert the same efficacy in the current study. The variations observed in these outcomes could be attributed to differences in experimental design since in our study, chicks were immunized SC on days one and seven of age and challenged with *Campylobacter* at day 14 of age, whereas Annamalai’s study involved immunization at one week and third week of age and *Campylobacter* challenge at day 35 of age. Another potential factor could be variations in the preparation methods of the OMPs. Nonetheless, co-administration of CpG ODN with *C. jejuni* OMPs significantly reduced *C. jejuni* counts by approximately 1.2 log_10_. When administered alone, CpG ODN reduced *Campylobacter* counts by approximately 1 log_10_, which aligns with our earlier observations [13]_._

While the reduction of *Campylobacter* counts was comparable between the group receiving CpG ODN alone and the group receiving both CpG ODN and *C. jejuni* OMPs, consistently higher levels of IgY and IgM antibody levels were observed solely in the latter group. However, consistent with our earlier observations [13], no correlation was noted in the groups with low *C. jejuni* CFUs and higher IgY and IgM Ab levels at the fifth week of age. It is worth noting that no differences in Ab levels were observed during the first two weeks of age, but increased levels were noted in the immunized groups starting from the third week of age and in the challenged group starting from the fourth week of age. This might be due to the absence of interfering effects of MDA on seroconversion following vaccination, as their levels significantly decreased by the second week of age and/or due to age-related changes in the immune system. These findings suggest that delaying the booster vaccination may be necessary to prevent potential interference from MDA with the antibody response and to achieve more efficient immune responses to the injected vaccine. Thus, in the subsequent study, we sought to investigate whether giving the booster doses of the vaccines at the second week of age would result in better outcomes.

To investigate the immunological mechanisms of protection further, the relative expression of key immune genes crucial in shaping the adaptive immune response was measured in the spleen and bursa of Fabricius of the immunized chickens. This included the T helper (Th)1-type cytokine (IFN)-γ, Th2-type cytokines (IL-4 and IL-13), proinflammatory cytokines (IL-1β), T regulatory cytokines (IL-10 and TGF-β), and B cell activating factor (BAFF) [50,51,52,53]. Varied expression levels of IFN-γ, IL-1β, and IL-13 cytokines were noted among the groups with lower *Campylobacter* colony counts. Specifically, the group immunized with CpG ODN showed elevated expression of IFN-γ, while the group immunized with the combination of CpG ODN and *C. jejuni* OMPs exhibited increased expression of IL-1β and IL-13. 

In addition to their proinflammatory role, induction of IFN-γ and IL-1β trigger the differentiating of naïve CD4+ T cells to Th1 and Th2 cells, respectively [26,54]. The increased expression of IFN-γ and IL-1β observed in this study aligns with our previous findings, where elevated expression of these genes was noted in the cecal tonsils and ileum following oral administration of CpG ODN, which correlated with a significant reduction in *Campylobacter* in broiler chicken [29]. In another study, an association was observed between the resolution of *S. Typhimurium* infection in experimentally infected hens and the increased expression of IFN-γ and IL-1β in the spleen and cecal tonsils [55]. Moreover, our previous in vitro study demonstrated the capacity of *C. jejuni* OMPs to induce robust immune responses in chicken macrophages and cecal tonsil mononuclear cells, including high expression levels of IFN-γ, IL-1β, and IL-13 [56].

Considering the role of IL-13 in the activation and differentiation of B cells and modulation of antibody-mediated immune responses against invading pathogens [52], the notable increase in the expression of IL-13 in the group receiving CpG ODN and *C. jejuni* OMPs may explain the enhanced IgM and IgY in this group and the associated reduction in *Campylobacter* counts. 

While CpG ODN and *C. jejuni* OMPs have shown the ability to induce immune responses and lower *Campylobacter* colonization when delivered SC, no such effects were observed when delivered in ovo and orally. The lack of consistent activity in these delivery methods could be ascribed to their chemical or mechanical degradation within the gastrointestinal tract [12]. These findings align with our previous research showing that administering soluble CpG ODN orally did not provide significant protection against *Campylobacter* [29]. However, encapsulating CpG ODN with PLGA nanoparticles was shown to enhance its immunostimulatory properties and protective effectiveness against *Campylobacter* colonization [13,57]. Hence, further investigations are needed to determine whether incorporating CpG ODN and *C. jejuni* OMPs into nanoparticles and administering them in ovo could improve their bioavailability at intestinal immune inductive sites, enhance mucosal and systemic immune responses, and provide protection against *Campylobacter* infection.

A second goal of this study was to optimize and evaluate the protective effects of various concentrations of *C. jejuni* OMPs and investigate whether altering the timing of the booster dose from the first week of age to the second week would lead to higher Ab production and better protection against *Campylobacter* colonization. We also deemed it worthwhile to examine the protective efficacy of multi-antigen vaccines, including heat-killed *Campylobacter* and whole-lysate vaccines, to identify the optimal antigens and dosages for future in ovo application. A significant reduction in *Campylobacter* counts was noted in the groups administered 50 or 200 µg of *C. jejuni* OMPs or a low dose of *C. jejuni* lysate. However, in line with the outcomes of the first study, no significant reduction in colony counts was observed in the group given 125 µg of *C. jejuni* OMPs. A significant increase in IgY antibody production was observed in the groups administered 50 µg of *C. jejuni* OMPs only at the third and fourth week of age. These results underscore the necessity of optimizing the dosage and timing of OMPs administration to enhance its effectiveness. 

Inactivated vaccines are known for their poor immunogenicity due to their limited ability to stimulate an adaptive immune response, making the inclusion of adjuvants necessary to enhance their effectiveness. In the context of their efficacy against *Campylobacter*, Glünder and colleagues observed a slight reduction in *Campylobacter* colonization in chickens vaccinated subcutaneously with formalin-inactivated *C. jejuni* and complete Freund’s adjuvant. Consistent with these observations, our results showed that SC immunization with heat-inactivated *Campylobacter* reduced the *Campylobacter* count by approximately 0.6 log_10_ [58]. In contrast, SC immunization with the whole lysate of *Campylobacter* resulted in a significant reduction in *C. jejuni* counts by approximately 1.4. These findings confirm and expand upon our earlier observations that orally administered *Campylobacter* lysate can reduce *Campylobacter* colonization [13].

Taken together, while *C. jejuni* lysate and OMPs have shown potential in reducing *Campylobacter* counts, the specific immunogenic protein responsible for these effects remains unclear. Additionally, it should be noted that their efficacy was evaluated in a homologous challenge model, and whether they demonstrate similar efficacy in a heterologous challenge model requires further investigation.

## 5. Conclusions

The findings of the present study suggest that vaccine formulations containing *C. jejuni* lysate or a combination of OMPs and CpG ODN show promise for reducing *C. jejuni* in broiler chickens. However, since the SC route is impractical for mass administration, further research is needed to determine whether using nanoparticles as a vaccine carrier could enhance their effectiveness in reducing *Campylobacter* colonization through more feasible routes, such as oral and in ovo administration.

## Figures and Tables

**Figure 1 vaccines-12-00908-f001:**
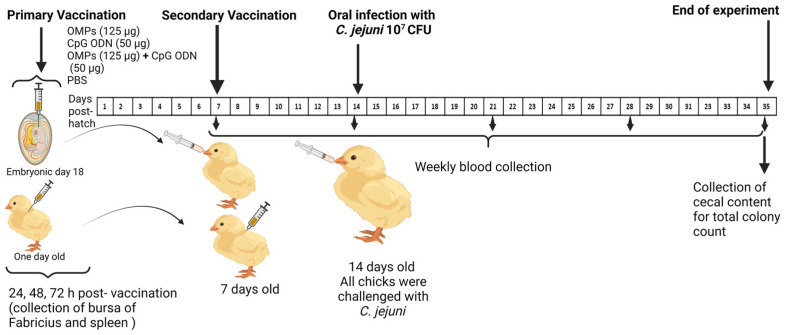
Illustration of first experimental design. On embryonic day 18, 136 eggs were randomly divided into four groups (G1–G4), each containing 34 eggs. Embryos of each group were injected intra-amniotic with the assigned immunization: 50 µg CpG ODN or 125 µg *C. jejuni* OMPs or their combination or PBS (negative control group). On the first day post-hatch, chicks (136) of non-immunized eggs were randomly allocated into four groups (G5–G8) and immunized SC with 50 µg CpG ODN or 125 µg *C. jejuni* OMPs or their combination or PBS. All chicks received the booster vaccination on day seven, and all groups were challenged with 10^7^ CFUs of *C. jejuni* on day 14. Bursa of Fabricius and spleens were collected for three successive days (*n* = 8) post-initial vaccination of either SC or in ovo immunized groups. Blood samples were collected weekly, and the cecal contents were collected on day 35 of age (the end of the experiment).

**Figure 2 vaccines-12-00908-f002:**
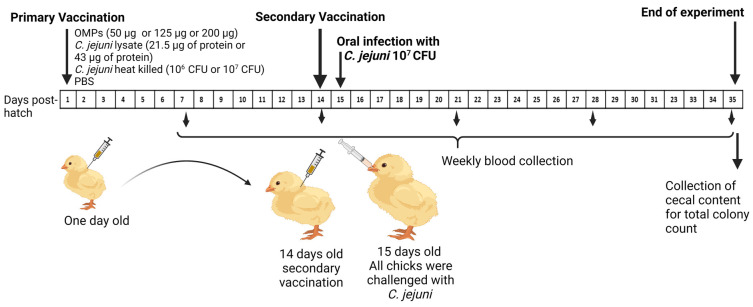
Illustration of the second experimental design. On the first day post-hatch, chicks were immunized SC with *C. jejuni* OMPs (50, 125 or 200 μg) or *C. jejuni* lysate; low (21.5 μg) or high (43 μg) protein, or heat-killed (10^6^ or 10^7^ CFUs of *C. jejuni*). All chicks received the booster vaccination on day 14 of age. The control group was injected at the same age with PBS only. All groups were challenged with 10^7^ CFUs of *C. jejuni* on day 15 of age. Blood samples were collected weekly, and the cecal contents were collected at the fifth week at the end of the experiment.

**Figure 3 vaccines-12-00908-f003:**
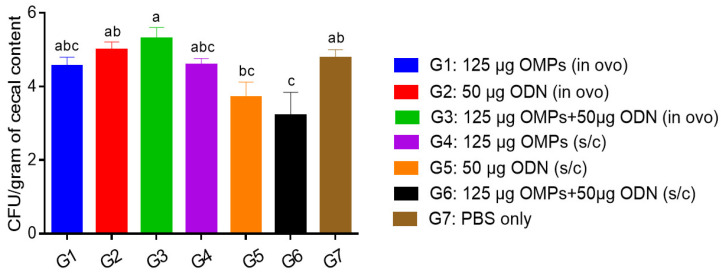
*C. jejuni* CFUs per gram of the cecal content. After primary immunization in ovo or SC on day one of age and booster immunization on day seven of age, chicks in all groups were orally challenged with 10^7^ CFUs of *C. jejuni* on day 14 of age. At 35 days of age (21 days post-challenge), cecal contents were collected for *Campylobacter* enumeration. Bars marked with different letters (a–c) indicate significant differences (*p* < 0.05) between the groups, while bars marked with the same letter denote no significant differences between the groups. OMPs = outer membrane proteins. SC = subcutaneous. ODN = synthetic single-stranded oligodeoxynucleotides (ODNs) containing unmethylated CpG motifs.

**Figure 4 vaccines-12-00908-f004:**
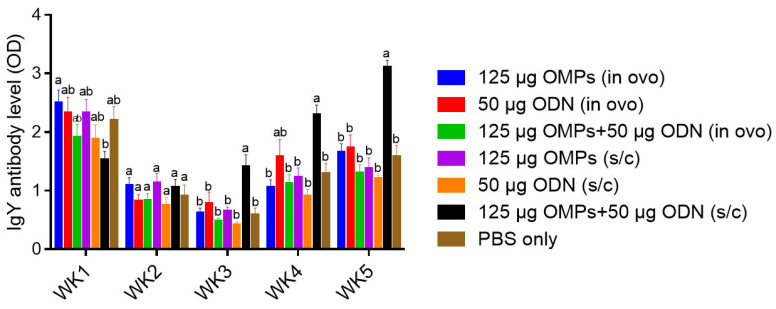
Serum IgY antibody levels. Chicks were immunized in ovo or SC with 125 µg *C. jejuni* OMPs or 50 µg CpG ODN or their combination or PBS. A booster dose was delivered orally (for those primed in ovo) or SC (for those primed SC) on day seven of age. All the chicks were then orally challenged with 10^7^ CFUs of *C. jejuni* on day 14 of age. Blood samples were collected weekly from all groups, starting from the first week of age through the fifth week of age, with the sera subsequently separated for measuring the IgY antibody (Ab) levels using ELISA. Bars marked with different letters (a–b) indicate significant differences (*p* < 0.05) between the groups, while bars marked with the same letter denote no significant differences between the groups. OMPs = outer membrane proteins. SC = subcutaneous. ODN = synthetic single-stranded oligodeoxynucleotides (ODNs) containing unmethylated CpG motifs.

**Figure 5 vaccines-12-00908-f005:**
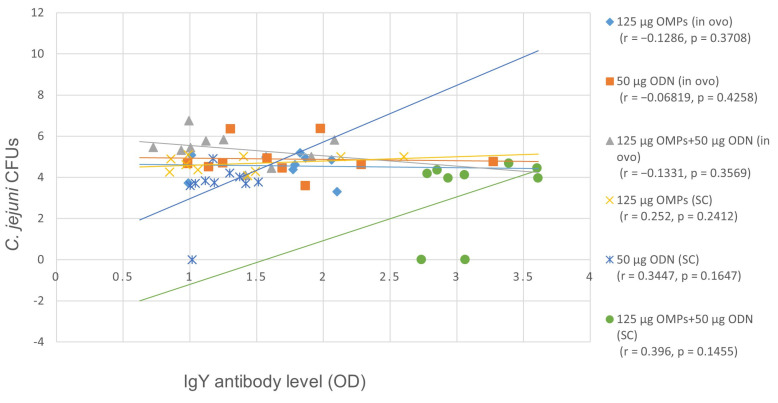
The correlation between *C. jejuni* CFUs and serum antibody (Ab) levels using Pearson’s r correlation coefficient. No correlation was observed between the serum IgY Ab levels and cecal counts of *C. jejuni* in the group immunized SC with the combination of 50 µg CpG ODN and 125 µg OMPs at the fifth week of age.

**Figure 6 vaccines-12-00908-f006:**
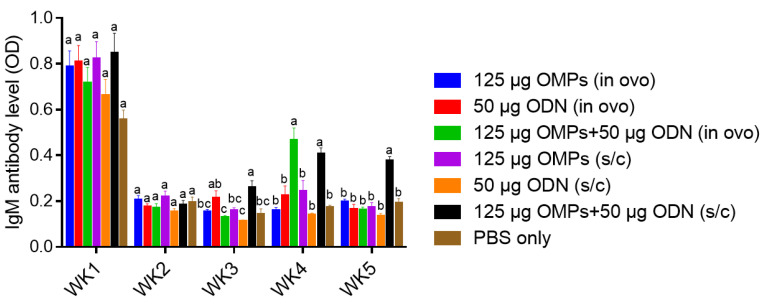
Serum IgM antibody levels. Chicks were immunized in ovo or SC with 125 µg *C. jejuni* OMPs or 50 µg CpG ODN or their combination or PBS. A booster dose was delivered orally (for those primed in ovo) or SC (for those primed SC) on day seven of age. All the chicks were then orally challenged with 10^7^ CFUs of *C. jejuni* on day 14 of age. Blood samples were collected weekly from all groups, starting from the first week of age through the fifth week of age, with the sera subsequently separated for measuring the IgM antibody (Ab) levels using ELISA. Bars marked with different letters (a–c) indicate significant differences (*p* < 0.05) between the groups, while bars marked with the same letter denote no significant differences between the groups. OMPs = outer membrane proteins. SC = subcutaneous. ODN = synthetic single-stranded oligodeoxynucleotides (ODNs) containing unmethylated CpG motifs.

**Figure 7 vaccines-12-00908-f007:**
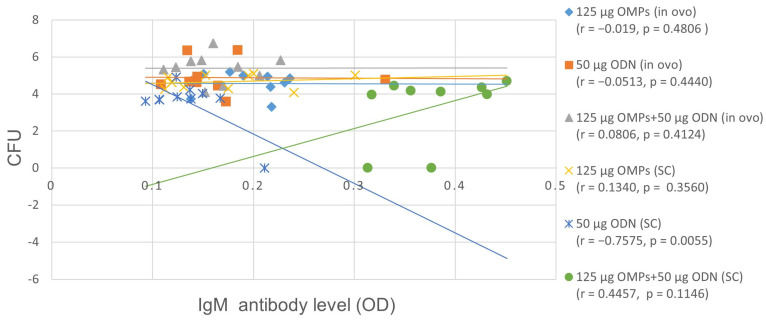
The correlation between *C. jejuni* CFUs and serum antibody (Ab) levels using Pearson’s r correlation coefficient. No correlation was observed between the serum IgM Ab levels and cecal counts of *C. jejuni* in the group immunized with the combination of 50 µg CpG ODN and 125 µg OMPs at the fifth week of age.

**Figure 8 vaccines-12-00908-f008:**
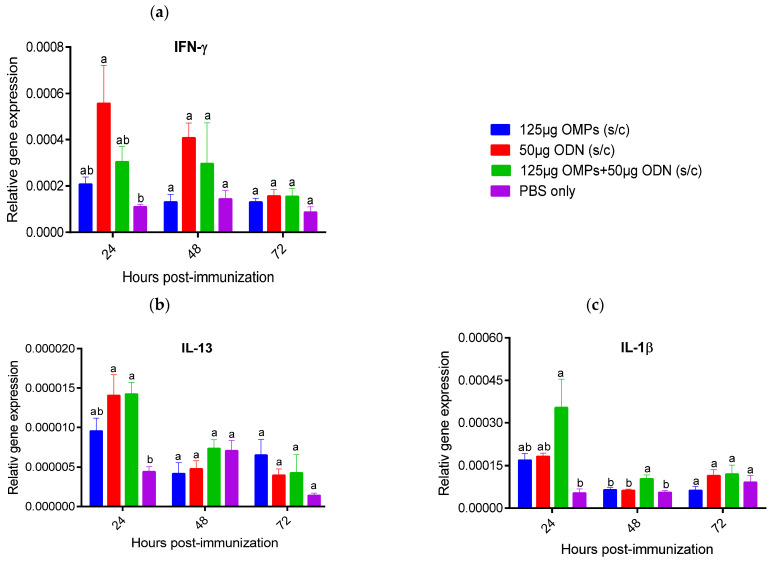
Relative gene expression of interferon (IFN)-γ (**a**), interleukin (IL)-13 (**b**), and IL-1β (**c**) in the spleen at 24, 48, and 72 h following SC immunization with 125 µg *C. jejuni* OMPs or 50 µg CpG ODN or their combination or PBS. Data are presented as mean expression (delta CT values) of cytokine mRNA relative to β-actin (housekeeping gene) ± standard error of the mean (SEM). Statistical significance among treatment groups was calculated using 1-way ANOVA followed by Tukey’s comparison test. Bars marked with different letters (**a**,**b**) indicate significant differences (*p* < 0.05) between the groups, while bars marked with the same letter denote no significant differences between the groups. OMPs = outer membrane proteins. SC = subcutaneously. ODN = synthetic single-stranded oligodeoxynucleotides (ODNs) containing unmethylated CpG motifs.

**Figure 9 vaccines-12-00908-f009:**
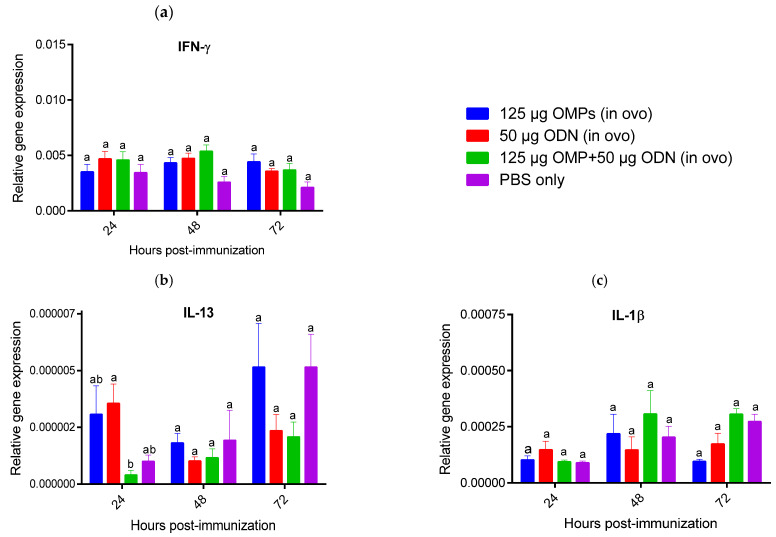
Relative gene expression of interferon (IFN)-γ (**a**), interleukin (IL)-13 (**b**), and IL-1β (**c**) in the spleen at 24, 48, and 72 h following in ovo immunization with 125 µg *C. jejuni* OMPs or 50 µg CpG ODN or their combination or PBS. Data are presented as mean expression (delta CT values) of cytokine mRNA relative to β-actin (housekeeping gene) ± standard error of the mean (SEM). Statistical significance among treatment groups was calculated using 1-way ANOVA followed by Tukey’s comparison test. Bars marked with different letters (**a**,**b**) indicate significant differences (*p* < 0.05) between the groups, while bars marked with the same letter denote no significant differences between the groups. OMPs = outer membrane proteins. SC = subcutaneously. ODN = synthetic single-stranded oligodeoxynucleotides (ODNs) containing unmethylated CpG motifs.

**Figure 10 vaccines-12-00908-f010:**
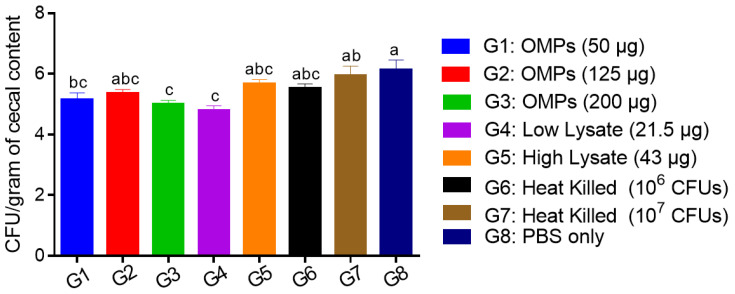
*C. jejuni* CFUs per gram of the cecal content. Day-old chicks were immunized SC with different doses of heat-killed or whole lysate or OMPs of *C. jejuni* and boosted SC with the corresponding vaccine on day 14 of age. On day 15 of age, chicks in all groups were orally challenged with 10^7^ CFUs of *C. jejuni*. On day 35 of age (20-day post-challenge), cecal contents were collected for *Campylobacter* enumeration. Bars marked with different letters (a–c) indicate significant differences (*p* < 0.05) between the groups, while bars marked with the same letter denote no significant differences between the groups. OMPs = outer membrane proteins. SC = subcutaneous.

**Figure 11 vaccines-12-00908-f011:**
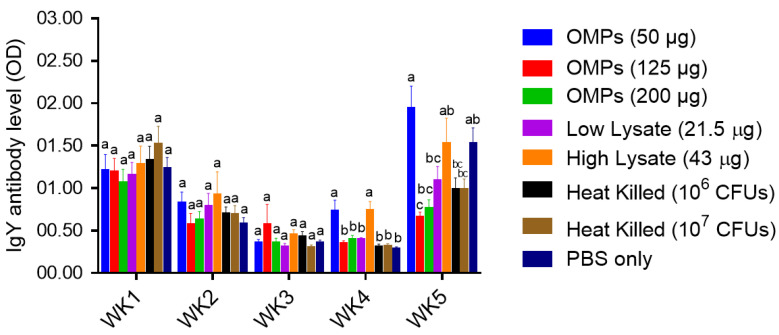
Serum IgY antibody levels. On days one and fourteen of age, chicks were immunized SC with different doses of *C. jejuni* OMPs (50, 125 or 200 μg) or *C. jejuni* lysate (21.5 μg or 43 μg) or heat-killed *C. jejuni* (10^6^ or 10^7^ CFUs). The control group was injected at the same age with PBS only. On day 15 of age, chickens were challenged orally with 10^7^ CFUs of *C. jejuni*. Blood samples were collected weekly from all groups, starting from the first week of age through the fifth week of age, with the sera subsequently separated for measuring the IgY antibody (Ab) levels using ELISA. Bars that are marked with different letters (a–c) indicate significant differences (*p* < 0.05). OMPs = outer membrane proteins. SC = subcutaneous.

**Figure 12 vaccines-12-00908-f012:**
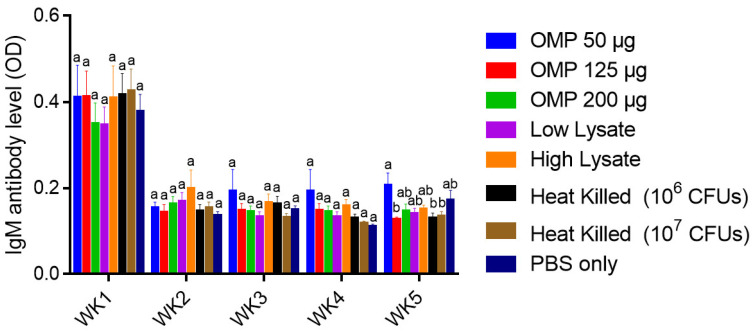
Serum IgM antibody levels. On days one and fourteen of age, chicks were immunized SC with different doses of *C. jejuni* OMPs (50, 125 or 200 μg), *C. jejuni* lysate (21.5 μg or 43 μg) and heat-killed *C. jejuni* (10^6^ or 10^7^ CFUs). The control group was injected at the same age with PBS only. On day 15 of age, chickens were challenged orally with 10^7^ CFUs of *C. jejuni*. Blood samples were collected weekly from all groups, starting from the first week of age through the fifth week of age, with the sera subsequently separated for measuring the IgM antibody (Ab) levels using ELISA. Bars marked with different letters (a–b) indicate significant differences (*p* < 0.05) between the groups, while bars marked with the same letter denote no significant differences between the groups. OMPs = outer membrane proteins. SC = subcutaneous.

**Table 1 vaccines-12-00908-t001:** Experimental design for the first trial.

	Treatment	Birds (*n*)/ Group	Primary Immunization			Booster Immunization	Challenge Age (d)
			Day	Route	Volume (mL)	Route/Volume (mL)/Age (d)	
1	*C. jejuni* OMPs (125 µg)	34	18th ED	Amniotic	0.1	Oral/1/7	14
2	CpG ODN (50 µg)	34	18th ED	Amniotic	0.1	Oral/1/7	14
3	*C. jejuni* OMPs (125 µg) + CpG ODN (50 µg)	34	18th ED	Amniotic	0.1	Oral/1/7	14
4	PBS	34	18th ED	Amniotic	0.1	Oral/1/7	14
5	*C. jejuni* OMPs (125 µg)	34	1-day old	SC	0.2	SC/0.2/7	14
6	CpG ODN (50 µg)	34	1-day old	SC	0.2	SC/0.2/7	14
7	*C. jejuni* OMPs (125 µg) + CpG ODN (50 µg)	34	1-day old	SC	0.2	SC/0.2/7	14
8	PBS	34	1-day old	SC	0.2	SC/0.2/7	14

ED = embryonic day, OMPs = outer membrane proteins of *C. jejuni*, SC = subcutaneous, *n* = sample size, CpG ODN = synthetic single-stranded oligodeoxynucleotides (ODNs) containing unmethylated CpG motifs.

**Table 2 vaccines-12-00908-t002:** Experimental design of the second trial.

Group	Treatment	Birds (*n*)/ Group	Primary Immunization			Booster Immunization	Challenge Age (d)
			Day	Route	Volume (mL)	Route/Volume (mL)/Age (d)	
1	*C. jejuni* OMPs (50 µg)	10	1-day old	SC	0.2	SC/0.2/14	15
2	*C. jejuni* OMPs (125 µg)	10	1-day old	SC	0.2	SC/0.2/14	15
3	*C. jejuni* OMPs (200 µg)	10	1-day old	SC	0.2	SC/0.2/14	15
4	*C. jejuni* heat-killed (10^6^ CFUs/bird)	10	1-day old	SC	0.2	SC/0.2/14	15
5	*C. jejuni* heat-killed (10^7^ CFUs/bird)	10	1-day old	SC	0.2	SC/0.2/14	15
6	*C. jejuni* lysate (21.5 µg)	10	1-day old	SC	0.2	SC/0.2/14	15
7	*C. jejuni* lysate (43 µg)	10	1-day old	SC	0.2	SC/0.2/14	15
8	PBS	10	1-day old	SC	0.2	SC/0.2/14	15

OMPs = outer membrane proteins of *C. jejuni, n* = sample size, SC = subcutaneous.

**Table 3 vaccines-12-00908-t003:** Sequences of primers used for RT-qPCR.

Gene	Primer Sequence (5′-3″)	Annealing Temp. (°C)	Reference
IFN-γ	F: ACACTGACAAGTCAAAGCCGC R: AGTCGTTCATCGGGAGCTTG	60	[28]
IL-10	F: TTTGGCTGCCAGTCTGTGTC R: CTCATCCATCTTCTCGAACGTC	64	[29]
IL-1β	F: GTGAGGCTCAACATTGCGCTGTA R: TGTCCAGGCGGTAGAAGATGAAG	64	[30]
IL-13	F: ACTTGTCCAAGCTGAAGCTGTC R: TCTTGCAGTCGGTCATGTTGTC	60	[31]
IL-4	F: GCTCTCAGTGCCGCTGATG R: GGAAACCTCTCCCTGGATGTC	58	[32]
BAFF	F: CACGTCATCCAGCAGAAGGAT R: ACAAGAGGACAGGAGCATTGC	55	[33]
TGF-β	F: CGGCCGACGATGAGTGGCTC R: CGGGGCCCATCTCACAGGGA	60	[34]
β-actin	F: CAACACAGTGCTGTCTGGTGGTA R: ATCGTACTCCTGCTTGCTGATCC	60	[28]

## Data Availability

The data presented in this study are available on request from the corresponding author.

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
