# Peer review of "Comparative Effectiveness of Various Multi-Antigen Vaccines in Controlling Campylobacter jejuni in Broiler Chickens"

_vaccines, 2024, doi:10.3390/vaccines12080908_

Round 1

Reviewer 1 Report

Comments and Suggestions for Authors

Reviewer's Comments:

  1. The study aims to evaluate the comparative effectiveness of various multi-antigen vaccines in controlling Campylobacter jejuni in broiler chickens. However, the introduction section lacks a clear motivation and justification for the choice of these specific vaccines and their combinations. Please elaborate on the rationale behind the selection of these vaccines.

  2. The results of the first trial indicate that subcutaneous immunization with the combination of CpG ODN and C. jejuni OMPs is effective in controlling Campylobacter colonization. However, the mechanism behind this efficacy is not fully discussed. Please provide insights into how the combination of CpG ODN and OMPs enhances the immune response and reduces Campylobacter counts.
  3. The second trial results comparing different antigen types are briefly mentioned but lack a detailed analysis and interpretation. Please expand on the comparison of the different vaccines and their effectiveness in controlling Campylobacter.

  4. The limitations of the study are not explicitly discussed. Please acknowledge any potential limitations in the experimental design, sample size, or other factors that may have influenced the results.

  5. The writing style could be improved by ensuring consistent terminology, avoiding repetition, and using more concise sentences. Please review the manuscript for clarity and conciseness.

Author Response

We appreciate your thorough review and valuable input. We have addressed all the concerns and incorporated the requested changes into the manuscript, which are highlighted in yellow.

Comment 1. The study aims to evaluate the comparative effectiveness of various multi-antigen vaccines in controlling Campylobacter jejuni in broiler chickens. However, the introduction section lacks a clear motivation and justification for the choice of these specific vaccines and their combinations. Please elaborate on the rationale behind the selection of these vaccines.

Response: We have significantly revised the introduction and provided a clear rationale for selecting these vaccines in our current study. Lines 54-87.

In this context, various multi-antigens vaccines, including killed, live attenuated, subunit, and recombinant vaccines, have been explored with differing levels of success [6,16]. Despite their multi-antigen composition, the ineffectiveness of these vaccines in providing adequate protection against Campylobacter was attributed to their lack of effective immunogenic proteins and the unsuccessful experimental setup, such as the use of suboptimal dosages and inefficient vaccination routes. Among these vaccines, the outer membrane proteins (OMPs) subunit vaccine has shown promise for controlling bacterial foodborne pathogens in chickens, including Salmonella and Campylobacter [11,12,17]. For instance, Annamali and colleagues have demonstrated that subcutaneous (SC) administration of a crude mixture of C. jejuni OMPs induced systemic protective antibody responses associated with a reduction in C. jejuni colonization in broiler chickens [11]. However, the SC route is impractical for mass administration in poultry production, necessitating further investigation into more suitable methods. While mucosal routes are appropriate for mass vaccination, oral delivery of vaccines remains challenging due to the potential degradation of vaccine antigens and adjuvants by gastric juice and digestive enzymes [13,14]. On the other hand, the in ovo route offers a reliable, cost-effective, and efficient approach for mass immunization, making it an attractive option for vaccinating thousands of birds at hatcheries, in contrast to laborious post-hatching vaccination procedures. Additionally, vaccine antigens delivered to chick embryos before hatching are not diluted by ingested food and have limited exposure to digestive enzymes, thereby prolonging their contact with mucosal tissues.

In contrast to live attenuated vaccines, it is widely agreed that inactivated and subunit vaccines require the incorporation of natural or synthetic adjuvants to enhance their immunogenicity. Among these adjuvants, CpG ODN, synthetic single-stranded oligodeoxynucleotides containing unmethylated CpG motifs, has demonstrated potential both as a standalone antimicrobial agent and as a vaccine adjuvant. CpG ODN is recognized by avian Toll-Like Receptor 21 (TLR21), which is analogous to TLR9 in mammals. When CpG ODN binds to TLR21, it triggers intracellular signaling pathways that stimulate the secretion of immunomodulatory substances like cytokines, chemokines, and antimicrobial peptides, which in turn enhance protection against bacterial pathogens [18,19]. Recent studies have shown that ovo administration of CpG ODN enhances chicks’ resistance to Escherichia coli and Salmonella Typhimurium infections [20]. However, the potential of in ovo-administered CpG ODN and C. jejuni OMPs to confer protection against C. jejuni has yet to be investigated.

Comment 2. The results of the first trial indicate that subcutaneous immunization with the combination of CpG ODN and C. jejuni OMPs is effective in controlling Campylobacter colonization. However, the mechanism behind this efficacy is not fully discussed. Please provide insights into how the combination of CpG ODN and OMPs enhances the immune response and reduces Campylobacter counts.

Response: We have added more details on the mechanisms behind the observed efficacy for CpG ODN and OMPs. Please see the changes highlighted in yellow on lines 546-571. To investigate the immunological mechanisms of protection further, the relative expression of key immune genes crucial in shaping the adaptive immune response was measured in the spleen and bursa of Fabricius of the immunized chickens. This included the T helper (Th)1-type cytokine (IFN)-γ, Th2-type cytokines (IL-4 and IL-13), proinflammatory cytokines (IL-1β), T regulatory cytokines (IL-10 and TGF-β) and B cell activating factor (BAFF) [48-51]. Varied expression levels of IFN-γ, IL-1β and IL-13 cytokines were noted among the groups with lower Campylobacter colony counts. Specifically, the group immunized with CpG ODN showed elevated expression of IFN-γ, while the group immunized with the combination of CpG ODN and C. jejuni OMPs exhibited increased expression of IL-1β and IL-13.

In addition to their proinflammatory role, induction of IFN-γ and IL-1β trigger the differentiating of naïve CD4+ T cells to Th1 and Th2 cells, respectively [23,52]. The increased expression of IFN-γ and IL-1β observed in this study aligns with our previous findings, where elevated expression of these genes was noted in the cecal tonsils and ileum following oral administration of CpG ODN, which correlated with a significant reduction in Campylobacter in broiler chicken [25]. In another study, an association was observed between the resolution of S. Typhimurium infection in experimentally infected hens and the increased expression of IFN-γ and IL-1β in the spleen and cecal tonsils [53]. Moreover, our previous in vitro study demonstrated the capacity of C. jejuni OMPs to induce robust immune responses in chicken macrophages and cecal tonsil mononuclear cells, including high expression levels of IFN-γ, IL-1β, and IL-13 [54].

Considering the role of IL-13 in the activation and differentiation of B cells and modulation of antibody-mediated immune responses against invading pathogens [50], the notable increase in the expression of IL-13 in the group receiving CpG ODN and C. jejuni OMPs may explain the enhanced IgM and IgY in this group and the associated reduction in Campylobacter counts.

Comment 3. The second trial results comparing different antigen types are briefly mentioned but lack a detailed analysis and interpretation. Please expand on the comparison of the different vaccines and their effectiveness in controlling Campylobacter.

Response: We have added the following section to the discussion (lines 600-610):

Inactivated vaccines are known for their poor immunogenicity due to their limited ability to stimulate an adaptive immune response, making the inclusion of adjuvants necessary to enhance their effectiveness. In the context of their efficacy against Campylobacter, Glünder and colleagues observed a slight reduction in Campylobacter colonization in chickens vaccinated subcutaneously with formalin-inactivated C. jejuni and complete Freund’s adjuvant. Consistent with these observations, our results showed that SC immunization with heat-inactivated Campylobacter reduced the Campylobacter count by approximately 0.6 log10. In contrast, SC immunization with the whole lysate of Campylobacter resulted in a significant reduction in C. jejuni counts by approximately 1.4 log10. These findings confirm and expand upon our earlier observations that orally administered Campylobacter lysate can reduce Campylobacter colonization.

We also welcome additional input from the reviewer on whether further discussion is needed.

Comment 4. The limitations of the study are not explicitly discussed. Please acknowledge any potential limitations in the experimental design, sample size, or other factors that may have influenced the results.

Response: We have added the following section to the discussion (lines 611-615):

“While C. jejuni lysate and OMPs appear to be promising vaccine candidates, the specific immunogenic protein responsible for these effects remains unclear. Additionally, it should be noted that their efficacy was evaluated in a homologous challenge model, and further investigation is needed to determine if they demonstrate similar efficacy in a heterologous challenge model.”

We have also modified our conclusion to the following (lines 618-623):

“The findings of the present studies suggest that vaccine formulations containing C. jejuni lysate or a combination of OMPs and CpG ODN show promise for controlling C. jejuni in broiler chickens. However, since the SC route is impractical for mass administration, further research is needed to determine whether using nanoparticles as a vaccine carrier could enhance their effectiveness in controlling Campylobacter colonization through more feasible routes, such as oral and in ovo administration.”

Comment 5. The writing style could be improved by ensuring consistent terminology, avoiding repetition, and using more concise sentences. Please review the manuscript for clarity and conciseness.

Response: As recommended by the reviewer, the manuscript underwent rigorous scrutiny, and significant changes were made to ensure clarity and conciseness.

Reviewer 2 Report

Comments and Suggestions for Authors

It's good to see studies investigating the potential for Campylobacter vaccination as such vaccines are likely to have a significant benefit for society if they can be shown to be effective. It's encouraging to see that some significant reductions in colonisation were seen in some groups. In vivo results, particularly involving Campylobacter, can be notoriously difficult to reproduce so it seems a pity that the second experiment didn’t reproduce and build on the most promising option from study 1 (125 OMP + ODN). In the discussion (L600-01) you have stated ‘whether giving the booster doses of the vaccines at the second week of age would result in better outcomes’ but there is only 1 group common to both studies (125 OMP) and in the first study this wasn’t protective and didn’t appear to induce a particularly strong antibody response. I think some comment/justification is needed as to why this group was chosen and why ODN (and/or another immunomodulator/adjuvant) wasn’t used.

For the cytokine work, why was expression only determined at 1-3d post 1st vaccination? Levels post-boost and post-challenge (possibly enabling correlation with colonisation level) would have been useful to know.

Other work that could have been considered: determining IgA responses, partic at the mucosal level, and characterising the antibody specificities (w. blots), particularly in birds that showed the most protection. Assuming this wasn’t done, it is worth discussing.

Comments on specific points in the paper:

L323 - ‘A positive correlation…between cfu and titer’ and L592 – ‘a positive correlation was noted in the groups with low C. jejuni CFUs and higher IgY and IgM Ab titers’.  A positive correlation is one where the variables move in the same direction. There’s one between ab level and protection, but an inverse correlation between antibodies and cfu. Or are you saying the birds with highest cfus had the highest ab levels? Also, strictly speaking your antibody results are not titers but absorbance (A405) readings. Suggest figures etc are relabelled as IgY ab ELISA reading (A405) etc and ‘antibody titers’ changed to ‘ab levels’. Also regarding the correlation work – was this done only with the wk5 samples or did it take into account levels from the earlier weeks (assuming that the birds were tagged so that earlier bleeds could be matched to cfus)? Some more explanation of this work is needed in the M&M.

Bar charts. It may be me, but I find the labelling of the bars regarding statistical differences unclear. ‘Bars that are marked with different letters (a–c) indicate significant differences (P < 0.05).’ I’m unclear on what this sentence means, and which groups differ from other groups. (Also, in some of the figures, there are no columns marked with a ‘c’.) Can you please rephrase or expand to make it clearer between which groups the signif diffs are.

L596. ‘This might be due to the absence of interfering effects…’ I think the increase in levels from wk3 may also be something to do with the fact there was effectively a 2nd boost @ wk 2 when the birds were challenged. There’s also a likelihood that the birds’ maturing immune systems play a role. Both of these need to be considered and discussed. An immature immune system is also one of the potential stumbling blocks for a successful Campy vaccine for broilers as their lives are normally very short – a point worth mentioning in the discussion.

Minor points:

L35 – ‘typically occurs….contact with farm animals..’ I’m not sure exposure via this route is ‘typical’ – livestock are obviously a risk but the majority of people have no contact with them. Suggest changing to something more generic, eg Handling and consumption of contaminated foods, with poultry products…

L46-47 – ‘Indeed, vaccination is considered essential..’ I’m not sure this is the case (unless you have a ref stating this?). A successful vaccine would certainly be of great benefit, but negative flocks can be and are produced by strict biosecurity alone, and there are other therapies that may have the potential to reduce or eliminate campy.

L89, 91, 120, 126, 272 (and possibly elsewhere) – replace microaerophilic (= the organism) with microaerobic (= the conditions)

L143 – replace ‘punched’ with ‘pierced’

L164 – more detail for the challenge: strain (I assume it was 81-176?), volume, buffer

L375-76. The relationship….ab titres. This is an incomplete sentence

L480. 1st day and 2nd week aren’t really ages. Suggest @ 1 d.o. and 7 (or 8?) d.o.

L508 – 3.2.2.1

L631 – ‘mechanical’??

Comments on the Quality of English Language

English is of good quality, barring the odd error (suggested corrections above)

Author Response

We appreciate your thorough review and valuable input. We have addressed all the concerns and incorporated the requested changes into the manuscript, which are highlighted in yellow.

Comment 1. It's good to see studies investigating the potential for Campylobacter vaccination, as such vaccines are likely to have a significant benefit for society if they can be shown to be effective. It's encouraging to see that some significant reductions in colonisation were seen in some groups. In vivo results, particularly involving Campylobacter, can be notoriously difficult to reproduce so it seems a pity that the second experiment didn’t reproduce and build on the most promising option from study 1 (125 OMP + ODN). In the discussion (L600-01) you have stated ‘whether giving the booster doses of the vaccines at the second week of age would result in better outcomes’ but there is only 1 group common to both studies (125 OMP) and in the first study this wasn’t protective and didn’t appear to induce a particularly strong antibody response. I think some comment/justification is needed as to why this group was chosen and why ODN (and/or another immunomodulator/adjuvant) wasn’t used.

Response: Thank you for your insightful comment. In the discussion, we clearly explained the rationale behind conducting the second study and established connections between the two studies. Additionally, we offered insights and recommendations for future directions in follow-up vaccine research. Please find our response to your concern below, along with the additional explanation highlighted in yellow in the discussion and conclusion sections.

Comment 2. For the cytokine work, why was expression only determined at 1-3d post 1st vaccination? Levels post-boost and post-challenge (possibly enabling correlation with colonisation level) would have been useful to know.

Response: We acknowledge the reviewer's suggestion to measure cytokines post-boost and post-challenge. We chose to measure cytokine expression at these specific time points to determine if early administration of CpG ODN to chick embryos and post-hatch could trigger an innate immune response in immature lymphoid organs, necessary for coordinating the adaptive immune response to the vaccine antigen. We did not assess immune responses post-boost due to our prior knowledge of its stimulatory effects at older ages. Previous studies showed oral CpG ODN administration in two-week-old chicks induced cytokine and antimicrobial peptide gene expression in the intestine and cecal tonsils. In addition, our recent research demonstrated that parenteral administration of CpG ODN at day 22 in broilers induced IL-2, IL-8, IL-10, and IFN-γ in the ileum (Abdelaziz’s unpublished data). As the reviewer knows, innate immune responses are non-specific and short-lived, so we did not expect these responses to persist until day 35 or to increase in magnitude if given on day 15. In our future work, we will involve measuring cytokine expression at later time points, considering the use of CpG encapsulated with nanoparticles for controlled release, which may result in a more sustained immune response. We hope that we have satisfactorily answered your questions.

Comment 3. Other work that could have been considered: determining IgA responses, partic at the mucosal level, and characterising the antibody specificities (w. blots), particularly in birds that showed the most protection. Assuming this wasn’t done, it is worth discussing.

Response: We recognize the important role of IgA in intestinal mucosal immunity, specifically against pathogens that proliferate within the intestinal lumen, such as C. jejuni. However, we were discouraged by the in-ovo and oral immunization data since none of the vaccine formulations was capable of reducing Campylobacter colonization or triggering cytokine expression and systemic antibody production. We also did not measure it in the subcutaneously immunized group as evidence indicates that parenteral vaccines do not effectively induce mucosal intestinal IgA responses. We will certainly consider measuring IgA in the intestine and bile in our upcoming study as we expect that nanoparticle encapsulation of CpG ODN and C. jejuni OMPs will enhance their efficacy when administered in ovo and orally, as previously reported for the orally administered PLGA-encapsulated CpG.

Comments on specific points in the paper:

Comment 4. L323 - ‘A positive correlation…between cfu and titer’ and L592 – ‘a positive correlation was noted in the groups with low C. jejuni CFUs and higher IgY and IgM Ab titers’. A positive correlation is one where the variables move in the same direction. There’s one between ab level and protection, but an inverse correlation between antibodies and cfu. Or are you saying the birds with the highest cfus had the highest ab levels? Also, strictly speaking, your antibody results are not titers but absorbance (A405) readings. Suggest figures etc are relabelled as IgY ab ELISA reading (A405) etc and ‘antibody titers’ changed to ‘ab levels’. Also, regarding the correlation work – was this done only with the wk5 samples, or did it take into account levels from the earlier weeks (assuming that the birds were tagged so that earlier bleeds could be matched to cfus)? Some more explanation of this work is needed in the M&M.

Response: We appreciate the time you took to critically review the manuscript. We agree that we measured antibody levels, not titers, replaced the word “titers” with “levels” throughout the manuscript, and relabeled the graphs. The correlation between CFUs and antibody levels was determined at the 5th week of age and colony count. We have clarified that no correlation was observed between IgY/IgM antibody levels and CFUs at the fifth week of age. The changes are highlighted in yellow.

Comment 5. Bar charts. It may be me, but I find the labeling of the bars regarding statistical differences unclear. ‘Bars that are marked with different letters (a–c) indicate significant differences (P < 0.05).’ I’m unclear on what this sentence means, and which groups differ from other groups. (Also, in some of the figures, there are no columns marked with a ‘c’.) Can you please rephrase or expand to make it clearer between which groups the signif diffs are.

Response: We appreciate your thorough review and apologize for any confusion. Our statement, "Bars marked with different letters (a–c) indicate significant differences (P < 0.05)," means that bars/groups marked with the same letter are not statistically different, while groups marked with different letters are statistically different. For example, if one group is marked with “a” and another group is marked with “b,” this indicates a statistically significant difference between them. Conversely, if one group is labeled “a” and another is labeled “ab,” this means there is no significant difference between these groups. As per the reviewer's suggestion, we have revised our phrase and expanded the sentence to the following: “Bars marked with different letters (a–b) indicate significant differences (P < 0.05) between the groups, while bars marked with the same letter denote no significant differences between the groups”

We have also corrected the letters in the figure caption to accurately reflect those on the bars.

Comment 6. L596. ‘This might be due to the absence of interfering effects…’ I think the increase in levels from wk3 may also be something to do with the fact there was effectively a 2nd boost @ wk 2 when the birds were challenged. There’s also a likelihood that the birds’ maturing immune systems play a role. Both of these need to be considered and discussed. An immature immune system is also one of the potential stumbling blocks for a successful Campy vaccine for broilers as their lives are normally very short – a point worth mentioning in the discussion.

Response: We have expanded this section and provided more clarification in the discussion (lines 537-541): “It is worth noting that no differences in Ab levels were observed during the first two weeks of age, but increased levels were noted in the immunized groups starting from the third week of age and in the challenged group starting from the fourth week of age. This might be due to the absence of interfering effects of MDA on seroconversion following vaccination, as their levels significantly decreased by the second week of age and/or due to age-related changes in the immune system. These findings suggest that delaying the booster vaccination may be necessary to prevent potential interference from MDA with the antibody response and to achieve more efficient immune responses to the injected vaccine. Please advise if any further expansion or clarification is required.

Minor points:

Comment 7. L35 – ‘typically occurs….contact with farm animals..’ I’m not sure exposure via this route is ‘typical’ – livestock are obviously a risk but the majority of people have no contact with them. Suggest changing to something more generic, eg Handling and consumption of contaminated foods, with poultry products

Response: Thank you for your suggestion. We have revised our statement to the following “Transmission of this organism to humans usually occurs through contact with farm animals or by consuming their contaminated products, with poultry products being the primary source of infection in most cases.”

Comment 8. L46-47 – ‘Indeed, vaccination is considered essential..’ I’m not sure this is the case (unless you have a ref stating this?). A successful vaccine would certainly be of great benefit, but negative flocks can be and are produced by strict biosecurity alone, and there are other therapies that may have the potential to reduce or eliminate campy.

Response: We agree with the reviewers that even with vaccinating chickens, other complementary measures are still required for the complete elimination of this pathogen. We have revised our statement to the following “Indeed, vaccination is considered one of the potential measures to control Campylobacter infection in chickens”

Comment 9. L89, 91, 120, 126, 272 (and possibly elsewhere) – replace microaerophilic (= the organism) with microaerobic (= the conditions)

Response: Thank you. We replaced microaerophilic with microaerobic throughout the manuscript.

Comment 10. L143 – replace ‘punched’ with ‘pierced’ Response: We replaced “punched” with “pierced”

Comment 11. L164 – more detail for the challenge: strain (I assume it was 81-176?), volume, buffer

Response: We have added more details in M&M. Lines 175-176: At two weeks of age, all groups were challenged orally with 107 CFUs of C. jejuni strain 81-176 in 1 ml PBS. Line 202: all groups were orally challenged with 107 CFUs of C. jejuni strain 81-176 in 1 ml PBS.

Comment 12. L375-76. The relationship….ab titres. This is an incomplete sentence

Response: Thank you, we have revised the whole section. And here is what we included in the revised version of the manuscript “A significant increase in IgM antibody levels was observed in the group immunized in ovo with both CpG ODN and C. jejuni OMPs only at the fourth week of age. Similar to the observation made for serum IgY, no correlation was observed between the groups with the low colony count and high IgM Ab titers at the fifth week of age (r = 0.4457, P = 0.1146) (Figure 7).

Comment 13. L480. 1st day and 2nd week aren’t really ages. Suggest @ 1 d.o. and 7 (or 8?) d.o.

Response: We have revised the sentence to the following : Immunizing chickens SC on days one and 14 of age with a low dose of C. jejuni lysate Comment 14. L508 – 3.2.2.1 Response: We added the missed number. Comment 15. L631 – ‘mechanical’?? Response: We mean mechanical degradation during the digestion of food in the gizzard.

Reviewer 3 Report

Comments and Suggestions for Authors

1. regarding the use of CpG, is it used separately from vaccine or mixed with vaccine? What is the detail of the procedure?

2. Why did CpG be used in the first experiment,while the adjuvant was not used in the second experiment? There are so many experimental variables, what is the purpose of each experiment? What conclusions can this experiment draw in the end, for the follow-up vaccine development? Need to explain?

3. Why choose in-egg inoculation? Is this approach feasible in actual production?

4. Figure 1 and Figure 2 are not clear, please use the original picture instead of the screenshot?

5. Why  only detect cytokines by mRNA? it is  too simple and one-sided to detect without flow cytometry?

6. It is necessary to perform autopsy and tissue section observation.

7. lack of ethical explanation in the article.

Comments on the Quality of English Language

Extensive editing of English language required

Author Response

We appreciate your thorough review and valuable input. We have addressed all the concerns and incorporated the requested changes into the manuscript, which are highlighted in yellow.

Comment 1. regarding the use of CpG, is it used separately from vaccine or mixed with vaccine? What is the detail of the procedure?

Response: We have provided more details in the material and methods Lines (152-158): “50 µg CpG ODN in 100 µl PBS or 125 µg C. jejuni OMPs in 100 µl PBS or their combination (50 µl of PBS containing 50 µg mixed thoroughly with 50 µl PBS containing 125 µg OMPs) were injected into the amniotic fluid of the fertilized eggs.” Lines (168-171): “one-day-old chicks were injected SC with 50 µg CpG ODN in 200 µl PBS or 125 µg C. jejuni OMPs in 200 µl PBS or their combination (100 µl of PBS containing 50 µg mixed thoroughly with 100 µl PBS containing 125 µg OMPs) or PBS.”

Comment 2. Why did CpG be used in the first experiment, while the adjuvant was not used in the second experiment? There are so many experimental variables, what is the purpose of each experiment? What conclusions can this experiment draw in the end, for the follow-up vaccine development? Need to explain?

Response: In the discussion, we clearly explained the rationale behind conducting the second study and established connections between the two studies. Additionally, we offered insights and recommendations for future directions in follow-up vaccine research. Please find our response to your concern below, along with the additional explanation highlighted in yellow in the discussion and conclusion sections. Our initial plan was to evaluate the combined effects of CpG ODN and C. jejuni OMPs, building on their demonstrated effectiveness in reducing Campylobacter counts when individually administered to chickens, as shown in previous research conducted by our group and others. However, the OMPs alone at a dose of 125 ug per egg/bird did not exert the same efficacy as those observed by Annamalai and colleagues. We attributed this inconsistency to the differences in experimental design since in our study, chicks have immunized SC on days one and seven of age and challenged with Campylobacter at day 14 of age, whereas Annamalai's study involved immunization at one week and third week of age and Campylobacter challenge at day 35 of age. Another potential factor could be variations in the preparation methods of the OMPs. On the other hand, the combination of CpG ODN and C. jejuni OMPs elevated interferon (IFN)-γ, interleukin (IL)-1β, and IL-13 gene expression in the spleen, significantly increased serum IgM and IgY antibody titers, and reduced cecal C. jejuni counts by approximately 1.2 log10. We, therefore, conducted the second study to optimize and evaluate the protective effects of various concentrations of C. jejuni OMPs. We also deemed it worthwhile to examine the protective efficacy of multi-antigen vaccines, including heat-killed Campylobacter and whole-lysate vaccines. This study provided valuable insights into identifying optimal antigens, dosage, and intervention age, paving the way for further research to enhance the efficacy of candidate antigens through the incorporation of adjuvants like CpG ODN or nanoparticle encapsulation for targeted delivery to mucosal inductive sites.

Comment 3. Why choose in-egg inoculation? Is this approach feasible in actual production?

Response: The reliability, cost-effectiveness, and suitability for mass administration make the in ovo route of vaccination an attractive and convenient immunization approach for vaccinating thousands of birds at hatcheries as opposed to laborious post-hatching vaccination procedures (Ricks et al. In ovo vaccination technology. Adv Vet Med. 1999;41:495-515). Indeed, in ovo vaccination is currently being used worldwide for protection against viral diseases, including Marek's disease (MD) and infectious bursal disease, with over 90% of commercial broiler chicks produced in the US are vaccinated in ovo against MD (Williams CJ, Zedek AS. Comparative field evaluations of in ovo applied technology. Poult Sci. 2010 Jan;89(1):189-93. doi: 10.3382/ps.2009-00093. PMID: 20008818.)

Comment 4. Figure 1 and Figure 2 are not clear, please use the original picture instead of the screenshot?

Response: We replaced the current figures with the original ones.

Comment 5. Why only detect cytokines by mRNA? it is too simple and one-sided to detect without flow cytometry?

Response: We acknowledge the reviewer's insights regarding the potential value of flow cytometry in uncovering the mechanisms of action of our vaccines at both cellular and molecular levels. Although flow cytometry was not utilized in this study, it is crucial to recognize that our research was primarily aimed at validating vaccine candidates, optimizing their dosages, administration routes and intervention age. Nevertheless, our findings have provided significant evidence supporting OMPs + CpG ODN and lysate as promising vaccine candidates, paving the way for further exploration into the potential enhancement of efficacy through nanoparticle encapsulation for oral and in ovo administration routes. Once the optimal routes and vaccine formulations are determined, deeper investigations into the vaccines' mechanisms of action at both cellular and molecular levels will be pursued.

Comment 6. It is necessary to perform autopsy and tissue section observation.

Response: We appreciate your suggestion to conduct an autopsy for tissue examination. However, it's important to note that C. jejuni is known as a commensal microorganism in the gut of chickens. Over the past 10 years, while working with C. jejuni strain 81-176 used in this study, we have not observed any abnormal clinical signs or pathological lesions in the intestines or other internal organs. In the first trial, we performed necropsies on eight embryos/hatched chicks per group for three consecutive days following in ovo and subcutaneous vaccinations for sample collection. During this process, we did not detect any pathological changes in the intestines or other internal organs. While we acknowledge the value of histopathological examination for revealing local immune responses, we did not perform it due to the absence of significant alterations in the expression of immune system genes or reductions in Campylobacter counts in orally vaccinated birds. We welcome additional input from the reviewer for consideration in our future investigations.

Comment 7. lack of ethical explanation in the article.

Response: We have provided the following statement in M&M “All procedures were approved by the Institutional Animal Care and Use Committee (IACUC) at Clemson University (AUP 2022-0411).

Reviewer 4 Report

Comments and Suggestions for Authors

The research results presented in the manuscript regarding the assessment of the effectiveness of vaccines against Camypylobacter jejuni in chickens are of great scientific value. The research was properly planned and carried out. The manuscript is well written.

However, I would like the authors to clarify a few issues.

L28-83 - Introduction – Are OMPs the most important antigens of C. jejuni when it comes to the production of protective antibodies? Are any other antigens being considered?

Have the authors considered the analysis of specific IgA on intestinal mucosa? This class of antibodies may play an important role in oral administration of the vaccine.

L72 - in ovo is Latin – consider italics throughout the manuscript

L620 – in vitro is also Latin

L86 - On what basis was the C. jejuni strain 81-176 selected for testing?

Chapter 2.2.2. – Did the authors check the content of the OMP preparation electrophoretically? If so, it is worth including a photo of the electrophoregram in the manuscript, at least in a supplement.

Chapter 2.2.3. and 2.2.4. – Were the inactivated vaccine and lysate prepared on the basis of the same strain as the OMP preparation?

Table 1 – explain please the abbreviation ED again under the table

L653 - The conclusion/summary should clearly refer to the results obtained following vaccination by different routes.

The manuscript should also emphasize that in large-scale poultry farming, the desired route of immunization is in ovo or per os.

L667 - Authors should include the number of the permit (document) for conducting experiments on animals.

Author Response

We appreciate your thorough review and valuable input. We have addressed all the concerns and incorporated the requested changes into the manuscript, which are highlighted in yellow.

The research results presented in the manuscript regarding the assessment of the effectiveness of vaccines against Camypylobacter jejuni in chickens are of great scientific value. The research was properly planned and carried out. The manuscript is well written.

However, I would like the authors to clarify a few issues.

Comment 1. L28-83 - Introduction – Are OMPs the most important antigens of C. jejuni when it comes to the production of protective antibodies? Are any other antigens being considered?

Response: To our knowledge, around 23 C. jejuni antigens have been identified and evaluated against Campylobacter. Among these, the cysteine ABC transporter substrate-binding protein (CjaA), Flagellin A protein, Campylobacter adhesion protein to fibronectin (CadF), and capsular polysaccharide (CPS) antigens were the most investigated. While these antigens have shown promise in reducing Campylobacter carriage, they lacked consistency, particularly against a heterologous challenge model. Therefore, research is ongoing to identify novel, highly conserved immunogenic proteins that can induce cross-protective immunity against different strains of C. jejuni. Our motivation to consider OMPs for this study stemmed from previous studies reporting their potential in controlling both Salmonella and Campylobacter. For instance, Annamalai and colleagues demonstrated that subcutaneous administration of a crude mixture of C. jejuni OMPs elicited systemic protective antibody responses, which correlated with a decrease in C. jejuni colonization in broiler chicks to below the detection limit. However, because the SC route is impractical for mass administration in poultry production, we conducted this study to investigate whether in ovo and oral vaccination of OMPs would yield similar outcomes, considering the suitability of these routes for mass vaccination. While in ovo and oral immunization with OMPs failed to enhance chickens' resistance against Campylobacter, our results for SC administration aligned with those of Annamalai and colleagues. Nonetheless, these results confirm the promising role of OMPs as vaccine antigens for reducing Campylobacter carriage in chickens. This research paves the way for further investigation into identifying the immunogenic/protective antigens within OMPs that were responsible for the observed effects in this study.

Comment 2. Have the authors considered the analysis of specific IgA on intestinal mucosa? This class of antibodies may play an important role in oral administration of the vaccine.

Response: We recognize the important role of IgA in intestinal mucosal immunity, specifically against pathogens that proliferate within the intestinal lumen, such as C. jejuni. However, we were discouraged by the in-ovo and oral immunization data since none of the vaccine formulations was capable of reducing Campylobacter colonization or triggering cytokine expression and systemic antibody production. We also did not measure it in the subcutaneously immunized group as evidence indicates that parenteral vaccines do not effectively induce mucosal intestinal IgA responses. We will certainly consider measuring IgA in the intestine and bile in our upcoming study as we expect that nanoparticle encapsulation of CpG ODN and C. jejuni OMPs will enhance their efficacy when administered in ovo and orally, as previously reported for the orally administered PLGA-encapsulated CpG.

Comment 3. L72 - in ovo is Latin – consider italics throughout the manuscript.

Response: While we value your feedback regarding italicizing "in ovo", we have looked over MDPI publications' guidelines and procedures, and we found out that "in ovo" is consistently used in MDPI articles without italics. We, therefore, have chosen to adhere to MDPI standard procedure and left "in ovo" unitalicized. We will also consult with the MDPI production team regarding the appropriate format during the production process if our manuscript is accepted for publication.

Comment 4. L620 – in vitro is also Latin.

Response: Please see the response above.

Comment 5. L86 - On what basis was the C. jejuni strain 81-176 selected for testing?

Response: C. jejuni 81-176 is a well-characterized and genetically stable strain. In addition, it effectively colonizes chicken intestines and has been used in numerous studies (Nothaft et al. Co-administration of the Campylobacter jejuni N-glycan-based vaccine with probiotics improves vaccine performance in broiler chickens. Appl. Environ. Microbiol. 2017, 83, e01523-17), (Taha-Abdelaziz et al. Oral administration of PLGA-encapsulated CpG ODN and Campylobacter jejuni lysate reduces cecal colonization by Campylobacter jejuni in chickens. Vaccine 2018, 36, 388–394), (Annamalai et al. Evaluation of nanoparticle-encapsulated outer membrane proteins for the control of Campylobacter jejuni colonization in chickens. Poult. Sci. 2013, 92, 2201–2211.)

Comment 6. Chapter 2.2.2. – Did the authors check the content of the OMP preparation electrophoretically? If so, it is worth including a photo of the electrophoregram in the manuscript, at least in a supplement.

Response: Please note that we utilized a standardized procedure for the extraction of outer membrane proteins (OMPs), which was previously validated using SDS-PAGE. The OMPs used in this study were also analyzed via SDS-PAGE and subsequently stained with Coomassie Blue (refer to the image below). We have incorporated the following sentence into the text: 'Protein separation was confirmed by SDS-PAGE and Coomassie Blue staining.

Comment 7. Chapter 2.2.3. and 2.2.4. – Were the inactivated vaccine and lysate prepared on the basis of the same strain as the OMP preparation? Response: Yes, C. jejuni strain 81-176 was used for the preparation of vaccine formulations in this work, including the inactivated, whole lysate and subunit OMPs vaccines.

Comment 8. Table 1 – explain please the abbreviation ED again under the table

Response: Thank you. We added ED = embryonic day.

Comment 9. L653 - The conclusion/summary should clearly refer to the results obtained following vaccination by different routes.

Response: Thank you for your valuable comment. We have modified our conclusion to the following (lines 618-623): “The findings of the present studies suggest that vaccine formulations containing C. jejuni lysate or a combination of OMPs and CpG ODN show promise for controlling C. jejuni in broiler chickens. However, since the SC route is impractical for mass administration, further research is needed to determine whether using nanoparticles as a vaccine carrier could enhance their effectiveness in controlling Campylobacter colonization through more feasible routes, such as oral and in ovo administration.”

Comment 10. The manuscript should also emphasize that in large-scale poultry farming, the desired route of immunization is in ovo or per os.

Response: We emphasized that oral and in ovo are the desired approaches for large-scale poultry (lines 620-623); “However, since the SC route is impractical for mass administration, further research is needed to determine whether using nanoparticles as a vaccine carrier could enhance their effectiveness in controlling Campylobacter colonization through more feasible routes, such as oral and in ovo administration”.

Comment 11. L667 - Authors should include the number of the permit (document) for conducting experiments on animals.

Response: Thank you. We have provided the following statement in M&M “All procedures were approved by the Institutional Animal Care and Use Committee (IACUC) at Clemson University (AUP 2022-0411).
